# Residue analysis evidence for wine enriched with vanilla consumed in Jerusalem on the eve of the Babylonian destruction in 586 BCE

**Ayala Amir** [1,2]*, **Israel Finkelstein** [1,3], **Yiftah Shalev** [4], **Joe Uziel** [4], **Ortal Chalaf** [4], **Liora Freud** [4], **Ronny Neumann** [2], **Yuval Gadot** [1]

**1** The Sonia and Marco Nadler Institute of Archaeology, Tel Aviv University, Tel Aviv-Yafo, Israel, **2** Department of Organic Chemistry, Weizmann Institute of Science, Rehovot, Israel, **3** School of Archaeology and Maritime Cultures, The University of Haifa, Mount Carmel, Haifa, Israel, **4** Israel Antiquities Authority, Jerusalem, Israel

* ayalaalbeck@gmail.com

**Data Availability Statement:** All relevant data are within the paper and its Supporting Information files.

## Abstract

The article presents results of residue analysis, based on Gas Chromatograph Mass Spectrometer (GC-MS) measurements, conducted on 13 ceramic storage jars unearthed in the Babylonian destruction layer (586 BCE) in Jerusalem. Five of the jars bear rosette stamp impressions on their handles, indicating that their content was related to the kingdom of Judah's royal economy. The identification of the original contents remains is significant for the understanding of many aspects related to the nutrition, economy and international trade in the ancient Levant. The study shed light on the contents of the jars and the destruction process of the buildings in which they were found. The jars were used alternatively for storing wine and olive oil. The wine was flavored with vanilla. These results attest to the wine consumption habits of the Judahite elite and echo Jerusalem's involvement in the trans-regional South Arabian trade of spices and other lucrative commodities on the eve of its destruction by Nebuchadnezzar.

## Introduction

During the 7[th] century BCE, Jerusalem enjoyed unprecedented prosperity, as it grew in size, population and wealth [1, 2]. The integration of Judah into the sphere of the Assyrian and later Egyptian empires, allowed the vassal kingdom to play an important role in the lucrative, long-range south Arabian trade, due to the fact that the main route of this network passed through the Negev [3–5]–the arid area in its southern sector. Several contemporaneously-composed biblical texts refer to the Arabian trade, but archaeology was yet to shed light on the commodities transported in this commercial system. The excavation of ceramic storage jars in the debris of the Babylonian (Nebuchadnezzar's) destruction of Jerusalem in 586 BCE presented us with an opportunity to examine the content of the vessels using Residue Analysis (RA). This is a key method for tracing ancient materials that are otherwise invisible in the archaeological record. It is based on the extraction, separation and identification of preserved and altered

**Funding:** The author(s) received no specific funding for this work.

**Competing interests:** The authors have declared that no competing interests exist.

biomolecular residues of the original contents from ceramic vessels, using GC/MS (gas chromatography/ mass spectrometry). Identifying such materials is vital for understanding the daily life and economy of ancient societies in matters such as dietary habits [6, 7], funerary and cultic customs [8, 9] and long-distance trade networks [10, 11].

The current study is based on two assemblages of jars that were found in storage rooms in two different locations in Jerusalem (Fig 1). The first is Building 100 in Area 10 in the Giv'ati Parking Lot excavations, on the southwestern slope of the Temple Mount. The assemblage is composed of at least 15 jars retrieved from a room belonging to a large public building that was destroyed during the Babylonian devastation of Jerusalem in 586 BCE [12, 13] (Figs 2 and S1 and S1 Table). Evidence for the destruction of the building includes massive debris of ash, burnt wooden beams and stone collapse from the second story. A study of soil samples and fragments of a plaster-made floor from the second story, found within the collapse, shows that the floor was exposed to a temperature of at least ˚600 C [13]. The ground story floor, where the storage vessels studied here were found, was exposed to a much lower temperature (see further below) (Fig 2).

The second assemblage comes from Structure 17049 in Area U, located on the eastern slope of the "City of David" ridge, to the south of the Temple Mount and above the Gihon Spring. The storage vessels were found in the middle room of this public structure, which was probably constructed in the 7th century BCE (Figs 3 and S2). Similar to Area 10, the room was packed with a thick destruction debris that included collapsed stones and a large quantity of burned pieces of wood–apparently belonging to the room's ceiling beams. Nine storage jars were restored and at least three additional jars are attested to by ceramic fragments [14].

Four of the restored storage jars from Area U and one from Area 10 bore handles with rosette stamp impressions, dated to the late 7th-early 6th century BCE. Rosette-impressed storage jars represent the royal distribution system of the kingdom of Judah on the eve of the Babylonian assault in 586 BCE [Fig 2A; 15; for paleomagnetic research on stamped handles, see, 16]. Other jars in Area 10 include smaller, bag shaped vessels [S1 Fig and ref. 17: Pl. 3.3.5] and a large pithos [ref. 18: Pl. 3.4.5: 2].

The location of some of the finds in a monumental structure on the slope of the Temple Mount, in close proximity to the Temple and palace of the Davidic kings, and the fact that some of the storage jars were stamped with a royal emblem, link the RA results to the elite circles in Jerusalem in the high days of its prosperity [e.g., ref. 19].

## Results

Identifying the original content of vessels using RA demands awareness of how diagenesis effects the way molecules are preserved [20]. Since the jars presented below were all found within the debris of buildings destroyed by violent conflagration, it is particularly challenging to assess the effect heating had on the preservation of different markers, especially wine-markers. In an experiment preformed on ceramic fragments enriched with wine, it was shown that heating causes changes in the relative proportions of the wine-markers [21]. These results will guide us while interpreting the results below.

Organic molecules reported here originated from the vessels, while they are absent from the control and blank samples (for the detailed results of each vessel, see S3 and S4 Figs and S2 and S3 Tables). All six jars from Building 100 contained tartaric acid and saturated fatty acids of 6–24 carbon atoms. In addition, they contained a variable composition of alcohols of 14–20 carbon atoms, sugar derivatives and monosaccharides derived from glucose, glycerol, lactic acid and a variety of fatty acids that include citric and malic acid derivatives, fumaric, glutaric, succinic and malonic acids. Three of the vessels (21748/5, 21748/1, 21748/4) also show peaks

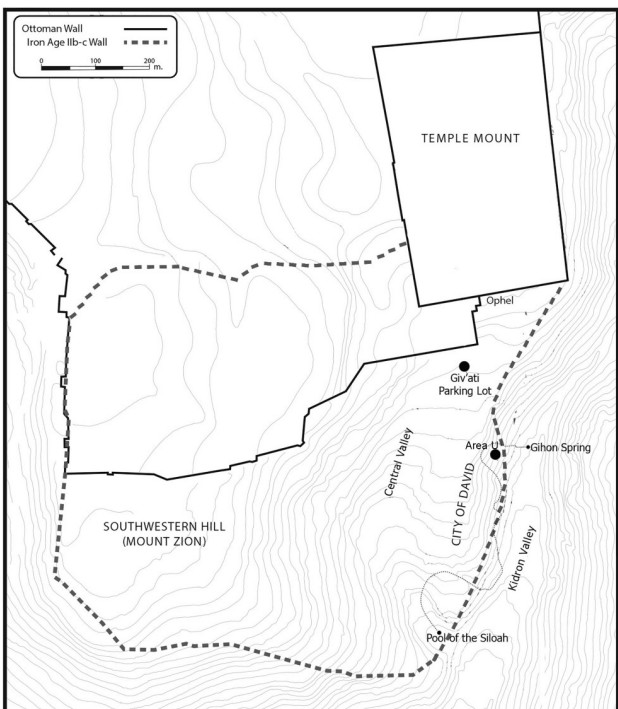

**Fig 1. Map showing the location of the two excavation area on the "City of David" ridge in Jerusalem.** Produced by Nitsan Shalom, the City of David expedition.

of vanillin. The pithos contained neither wine-markers nor vanillin, but saturated fatty acids of 16–24 carbon atoms and oleic acid and therefore serves as a control, validating the reliability of the results obtained for the storage jars (Figs 4 and S3). Four jars from Structure 17049 underwent the two extraction steps (the others were not analyzed by the wine-markers extraction [WM hereon] due to poor results from the total lipid extraction [TLE hereon]). Two were

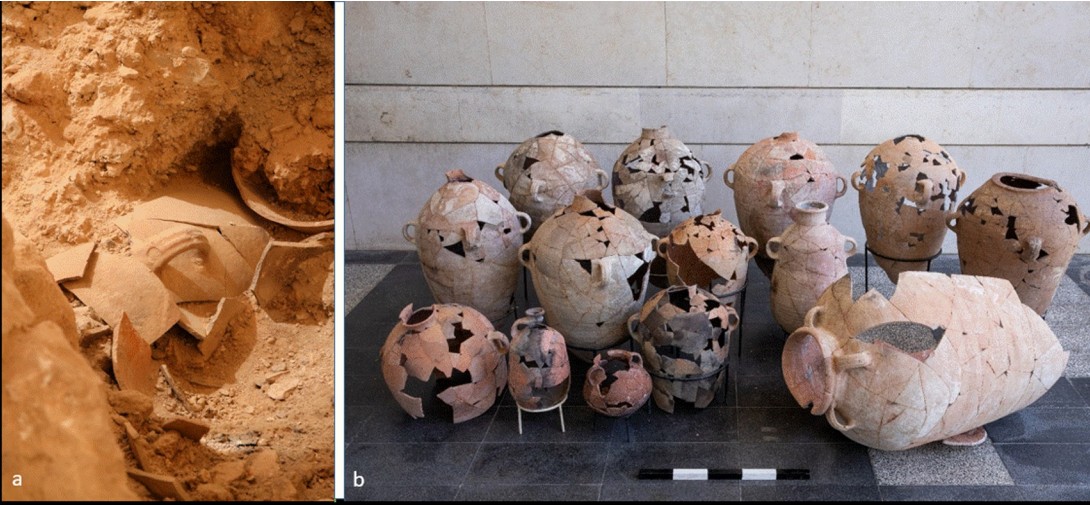

**Fig 2. Vessels found in Room C, Building 100.** a. The smashed jars while being excavated inside Room C. b. The assemblage of storage jars found inside Room C following restoration. Photographed by Sasha Flit, Tel-Aviv University.

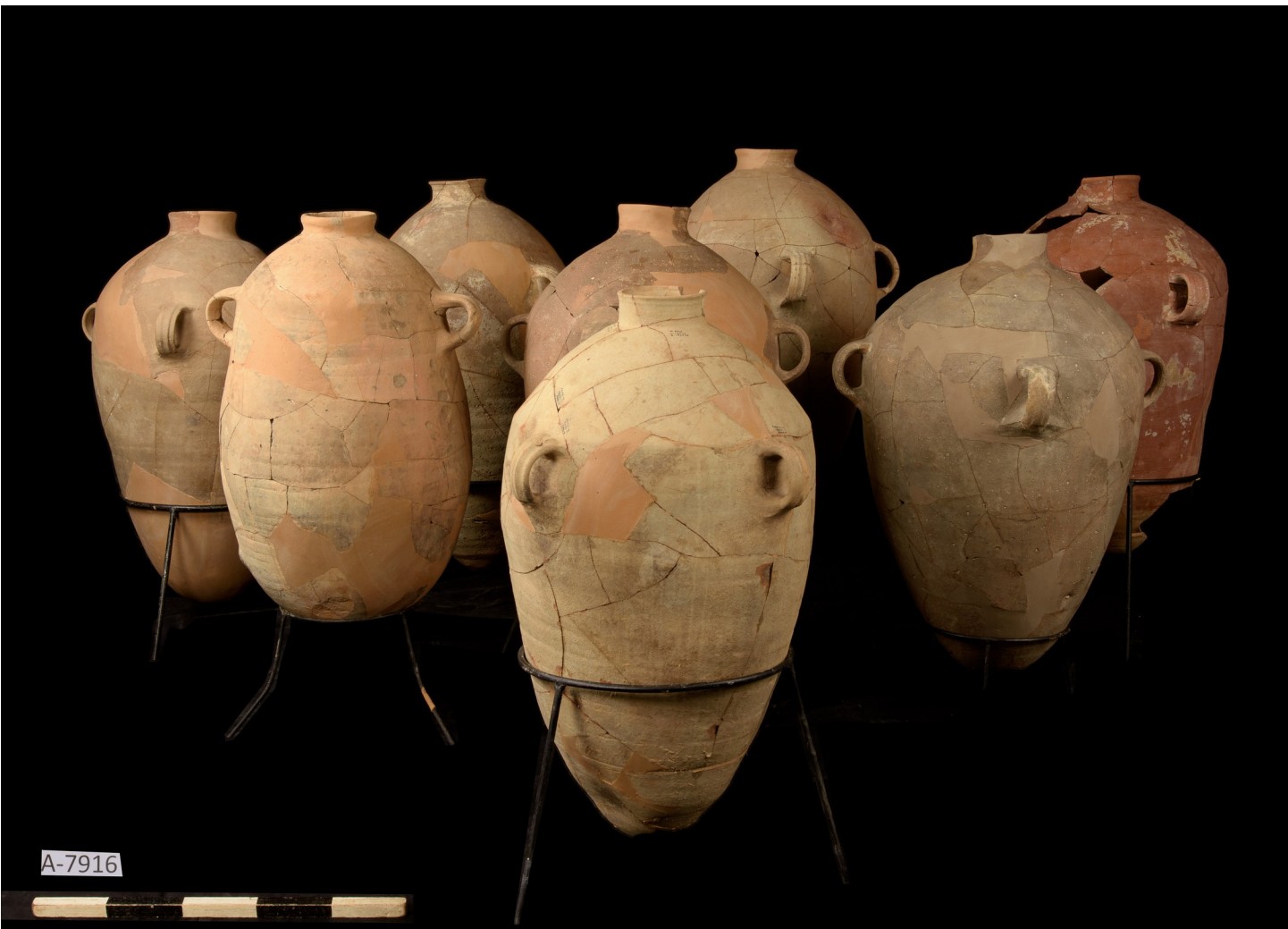

**Fig 3. The assemblage of storage jars found within structure 17049.** Photographed by Dafna Gazit, the Israel Antiquities Authority (IAA).

rich with organic material. Vessel 170483 contained palmitic, stearic and oleic acids in the relative ratio of $C_{16:0} > C_{18:1;9} > C_{18:0}$, and also myristic ($C_{14:0}$), linoleic ($C_{18:2; 9,12}$), and arachidic ($C_{20:0}$) acids. This vessel also contained tartaric acid, glycerol, citric and malic acid derivatives, fumaric, glutaric, succinic and malonic acids, as well as $MAG_{16:0}$, $MAG_{18:0}$, n-alkanes of 23–29 carbon atoms and alcohols of 12–18 carbon atoms. Vessel 170571 contained maleic, succinic glutaric, and fumaric acids, glycerol, alcohols of 12–20 carbon atoms and saturated fatty acids. It did not contain tartaric acid. Both vessels contained vanillin (4-hydroxy-3-methoxybenzaldehyde), 4-hydroxybenzaldehyde, and acetovanillone (Figs 5 and S4).

## Discussion

### Wine

A variety of molecules were identified in both jar assemblages, though their composition varies slightly between the vessels. The marked difference between the results obtained in the jars and those of the control samples indicates that the former indeed reflects the jar's original

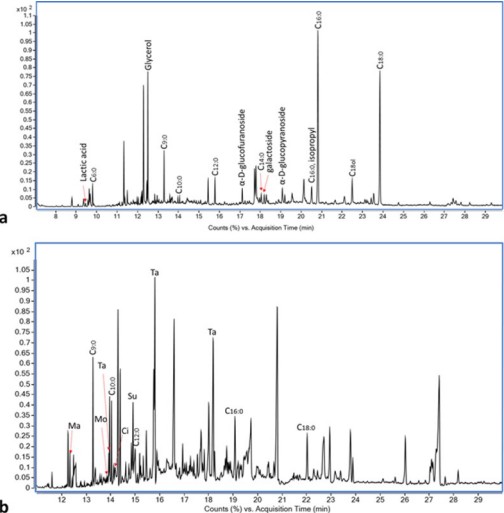

**Fig 4. Chromatogram of vessel #21674/2 from Building 100.** a. Partial TLE Chromatogram of vessel #21674/2. b. Partial WM Chromatogram of vessel #21674/2. Cx:y = fatty acid with x carbons in its chain and y is the number of double bonds; Cxol = alcohol with x carbons in its chain; Ma = malic acid; Mo = malonic acid; Ta = tartaric acid; Ci = citric acid; Su = succinic acid. Derivatization information was omitted for clarity. For details see S3 Fig.

contents. Tartaric acid is a principal biomarker of grape [22, 23]. Other molecules also appear in wine; these include fumaric and glutaric acids that originate from the grape. Succinic, maleic and malonic acids are acids produced by fermentation of the grape juice [24]. The combination of these acids—together with glycerol, lactic and tartaric acids—attest to the content of wine in these jars.

Syringic acid, which is also a wine-marker that appears in red wine, was not detected in the jars, although two (170483 and 170571) were extracted by an alkaline treatment developed by Pecci et al (2013) [21] in addition to the TLE and WM extractions described below. Still,

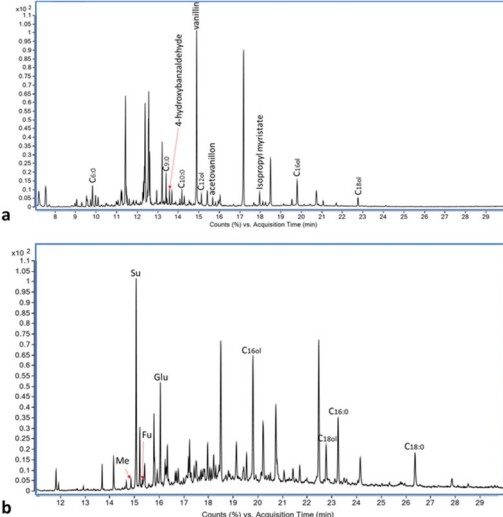

**Fig 5. Chromatogram of vessel #170571 from structure 17049.** a. Partial TLE Chromatogram of vessel #170571. b. Partial WM Chromatogram of vessel #170571. Cx:y = fatty acid with x carbons in its chain and y is the number of double bonds; Cxol = alcohol with x carbons in its chain; Me = maleic acid; Su = succinic acid; Fu = fumaric acid; Glu = glutaric acid. Derivatization information was omitted for clarity. For details see S3 Fig.

although alkaline treatment is the optimal method for the syringic acid extraction, its absence is not enough to indicate white wine in the jars [25, 26]. The monosaccharides identified in many of the jars may have originated from natural sugars of the wine, or rather from wine additives that sweetened it.

It seems, then, that all six jars examined from Building 100 and at least two jars from Structure 17049 (170571 and 170483) contained wine. The wine-marker profiles of most samples had specific repetitive characteristics in terms of the relative abundance of the wine-associated fatty acids. Tartaric acid is the most abundant, followed by succinic acid and then by the other fatty acids: fumaric, malic, malonic, maleic, glutaric and citric acids. One of the samples from the base of Vessel 170483 had a profile in which malic acid predominates the other acids.

The profile of one storage jar from Structure 17049 (# 170571) is different from the others. Tartaric acid is absent and succinic acid is the most dominant peak, followed by maleic, fumaric and glutaric acids. Such changes in the relative proportions of these molecules were demonstrated by Pecci et al. (2013) to be the result of degradation caused by heating. An experiment in which a fragment of a pan containing cooked wine was kept in an oven at 70˚ C for eighteen days, showed that the heating process reduced the relative abundance of the tartaric acid, whereas succinic acid became the most abundant in the sample [21]. It seems that while most of our wine profiles are consistent with those obtained from Pecci's archaeological samples and experimental sample of buried ceramic, the profile of Jar 170571 is more apt to the heated sample.

Since all the storage jars under examination were found within the destruction, we must ask why only one storage jar shows evidence for the wine having been exposed to heat? The most probable interpretation is that Jar 170571 was exposed to a higher temperature than all other jars. Indeed, the exposure to heat in Structure 17049 was not even, as attested by soot found only on some of the sherds. It may be that Jar 170483 was located farther away, or hidden under the debris during the conflagration. A paleomagnetic study of some of the ceramics from both assemblages showed that they did not record the magnetic north during the destruction of the room, but rather kept the magnetic signal that they acquired in the kiln, meaning that they were not exposed to high temperature. An analysis conducted on fragments of well-made plaster that originally served as the upper story floor, indicates that it was exposed to high temperature. At the same time samples of earth taken from the immediate surrounding of the storage jars show that they were not exposed to high temperature. All this indicates that the main conflagration in Building 100 occurred in the upper story which then collapsed into the lower story, where the storage jars were standing [13].

Two more possible explanations for this discrepancy seem to be less plausible. One is that the seven jars with a standard signal of wine were smashed by the falling debris before they caught fire. Since the wine molecules were trapped at the time of the fire within the pores of the clay, their composition would be altered by the heat even if the liquid had spilled. A second explanation is that the heating process occurred prior to storage of the wine in the jar, a deliberate warming and cooking of the wine with sugar and spices aimed at preparation of mulled wine. This possibility seems less likely as there is no known archaeological or textual evidence for this practice during the period discussed here and as heating of wine was usually done in smaller serving vessels.

## Other markers

**Vanilla residues.** The most surprising results in this study are the profiles obtained from the two jars from Structure 17049 and three jars from Building 100, indicating the presence of vanillin. Vanillin can be found in grapes and wine, but in low concentrations [27–29]. It further appears in minor amounts in aromatic and balsamic materials, such as benzoe and storax

resins and in rosemary (*Rosmarinus officinialis*). These materials contain other major components and biomarkers that were not identified in the jars from Jerusalem [30–32]. The profiles obtained from Jars 170571 and 170483 are similar to those observed in the Megiddo juglets analyzed by Linares et al., who demonstrated that Vanilla was imported to the Levant during the Middle Bronze Age III (ca. 1600 BCE), long before it was domesticated in the New World [10]. The dominance of vanillin in the jars, the appearance of acetovanillon, and the absence of other major components of substances that contain vanillin, all testify to the presence of vanilla in the jars, rather than resins or other materials. Unlike Jars 170483, 170571 from Structure 17049, the three jars from Building 100 did not contain 4-hydroxybenzaldehyde and acetovanillone, and their vanillin peak was relatively smaller. This may indicate a lower level of preservation of the vanilla markers in these jars. The vanillin alone is not sufficient to determine its source. The facts that Jars 170483 and 170571 from the adjacent wareroom contained vanilla markers and the above-mentioned three jars contained neither biomarkers of other potential sources of vanillin, reinforce the possibility that these jars contained vanilla as well.

The large capacity of the jars and the presence of wine-markers may indicate that the vanilla was used as a wine additive. Archaeological evidence for flavoring wine with exotic spices has recently begun to accumulate [32–34]. DNA analysis conducted on several Aegean amphora dated to the 5th-3rd century BCE revealed remains of both grapes and herbs at the same vessels. It is unclear however if these are the remains of the same content (i.e spiced wine) or of two separate ones [35]. Mapping the possible sources of vanilla cords imply that they were imported from either India or east Africa [10]. Both areas were connected to the Levant by the desert roads which originated ether in South Arabia or Egypt. Archaeological and textual evidence show that the Southern Arabian trade network flourished throughout the 7th century BCE, first under the Assyrian empire and later under their successors, the Egyptian 26th Dynasty and Babylonia. The main Arabian trade route to the northwest passed in the Beer-sheba Valley in the territory of Judah, and the kingdom probably provided shelters and supplies to the convoys. Finding vanillin in Jerusalem is an indication that the city was one of the destination of some of the elite products which were transported from Arabia. This is supported by earlier finds, such as the three inscribed sherds written in South Arabian script found in the "City of David" Jerusalem excavations [36] and in Tel ʿAroer in the Beer-sheba Valley [37], as well as by the Sabaean inscription from ca. 600 BCE, referring to the "towns of Judah" [1]. This trade system is also mentioned in the biblical texts, especially in the second book of Kings and the book of Jeremiah (e.g., "To what purpose cometh there to me incense from Sheba, and the sweet cane from a far country?", Jer. 6:20). The story of the visit of the Queen of Sheba to Jerusalem, bringing exotic gifts to Solomon (I Kings 10:2), may reflect realities of the 7th century BCE–the incorporation of Judah into the Assyrian-led Arabian trade [38]. Despite exegesis complexities, these texts show that during the last days of the kingdom of Judah, its elite enjoyed fragrances and spices which were imported from afar, probably using them within the cultic as well as daily realms. The identity of these products unrecognized so far, should be re-evaluated in light of the finds described here.

**Olive oil.** In addition to wine-markers, Jar 170483 contained biomarkers of olive oil. This identification is based on the presence of palmitic, oleic, stearic and arachidic acids, in which the oleic acid peak predominates the stearic acid peak [39–41]. Other jars may have been used to contain olive oil as well: myristic, palmitic and nonanoic acids, which appear in many jars, as well as octadecanol, may have originated from olive oil [42–44]. It seems therefore that at least some of the jars studies here were used for storage at least twice: for olive oil and for wine intermittently.

That storage vessels were multi-purpose and not pre-designated for the storage of a single product is well attested, for instance in a multidisciplinary study of graffiti and inscriptions

found on Aegean Amphorae [45–48], coupled with RA and DNA analysis of their contents [e.g., refs. 35, 49, 50]. Jars bearing rosette stamped handles began to be produced as early as ca. 620 BCE [15]. Furthermore, specific measurements of the intensity of the magnetic field performed on both floor plaster segments that were heated in the destruction found in the collapse of Building 100 and the jars prove that storage jars stamped with rosette impressions were manufactured several decades before the destruction [13, 16]. This means that these jars were kept in use for a long period, probably reused for different purposes. This is an important addition to the study of the royal administration of Judah in the period when jars were stamped–from the late 8[th] to the early 6[th] century BCE.

**Sealing.**    Jar 170483 contained n-alkanes of 23–29 carbon atoms and $MAG_{16:0}$, $MAG_{18:0}$ as well. The hydrocarbons also occur in Jar 21748/1 found in Building 100. Odd- numbered n-alkanes of 23–31 carbon atoms, of which $C_{29}$ is the major peak, appear in other jars from Structure 17049. These molecules occur widely in many waxes of animal and vegetal sources such as epicuticular wax and beeswax, and could be applied to the vessels as a sealant [8, 51, 52]. The fact that the rosette jars were used for the storage of wine reinforces the need for sealing and/or coating the vessels in order to prevent the oxidation and spoilage [32, 44]. A group of three stoppers found alongside the jars in Structure 17049 provide another indication for actions taken in order to prevent the entrance of air [34, 53].

## Conclusions

In this article we presented the first archaeological evidence for the contents of storage jars bearing stamped impressions that served the royal Judahite administrative system [54]. The results shed light on wine consumption habits of the elite circles in the capital and on Judah's involvement in long distance trade networks. Apparently the jars were used for the storage of olive oil and wine–the two typical products of the kingdom under Assyrian domination [55], and were sealed to avoid oxidation of their contents. The fact that the jars were recycled reinforces the assertion that the manufacturing of Jars stamped with Rosette stamp impressions began a few decades before the Babylonian siege and not as part of the preparation for that siege [56].

Residues of vanilla, discovered in some of the jars, attest to the great prestige of the wine and to the drinking habits of the elite residents of Jerusalem. Vanilla had to be imported from the tropic environments of India or east Africa. Control over the spice trade routes connecting east and west has often been seen as a prime motivator for the Assyrian expansion to the southwest. The identification of vanilla as one such exotic and prestigious product having been brought over by the desert caravans highlights the economic value of this trade. We demonstrate that vanilla used as a wine additive by the kings of Judah and their entourage. The royal elite of the kingdom, residents of Jerusalem, webbed into this trading network, serving as clients of the Assyrian and later Egyptian empires.

Last but not least, we presented novel evidence for the effect of heating on the way residues of wine were preserved in ancient ceramic vessels. It is challenging to explain why we see these changes in only one storage jar, while the other seven (from two different locations) show markers for unheated wine. Based on the available evidence, which include analysis of soil samples and reconstruction of the direction and intensity of the magnetic field, we assume that most of the storage jars were not exposed to the direct heat of the fire as the buildings collapsed.

## Materials and methods

### Materials

Analyses were performed on six jars and one pithos found in Building 100 from Area A (S1 Fig and S1 Table), and ten jars found in Structure 17049 in Area U (S2 Fig). The samples were

taken from the vessel's base. From Structure 17049, one jar (#170571) was sampled at the neck and base, and another (#170483) at the body and base. Fresh samples of sediment to be used as controls, were taken from inside each vessel and from its immediate vicinity. [57].

## Methods

The extractions were carried out in two steps. The first step was the total lipid extraction (TLE), followed by the second step which was the wine-markers extraction (WM).

**Total lipid extraction.**    The extraction and analysis procedures of the lipids from the ceramic vessels followed Evershed et al. (1990) [58] and Charters et al. (1993) [59]. All glassware were pre-heated at 100˚C oven. 1g fragments were broken off the ceramic vessels using nitrile gloves and clean pliers, ground to a powder in an agate mortar and pestle and transferred into 8 ml vials. 5.0 ml of dichloromethane and methanol (2:1, v:v) were added to each vial and the mixture was sonicated at 80˚C for 15 min. The vials were centrifuged for 10 min at 3500 rpm and the supernatant was transferred to another clean vial. The extraction steps were repeated three times. The solvents were then evaporated under a gentle stream of nitrogen and mild heating.

**Wine markers extraction.**    The extraction and analysis procedures of the wine-markers from the ceramic vessels followed Garnier and Valamoti (2016) [60]. 5.0 ml of boron trifluoride, butanol and cyclohexane (1:2:4, v:v) were added to each vial of the powder remaining from the TLE extraction. The mixture was sonicated at 80˚C for 2 hours, and then it was neutralized by an aqueous saturated solution of sodium carbonate. 2.0 ml of dichloromethane were added before vortexing it for 1 min. and centrifuging it for 10 min at 3500 rpm. The supernatant was transferred to another clean vial. The DCM addition, vortex and centrifugation steps were repeated twice. 2.0 ml of distilled water were then added, and the mixture was vortexed for 1 min. and centrifuged for 10 min at 3500 rpm. Two phases were created, where the upper phase contained the water which was removed from the sample. The washing steps were repeated once more. Anhydrous sodium sulfate was added to the vials in order to dry the samples from water that may have been left in them. Then the solution was removed to a clean vial and evaporated to dryness under a gentle stream of nitrogen and mild heating.

**Derivatization by silylation.**    To the TLE samples: 100.0 μl of N,O-bis(trimethyl)silyltrifluoroacetamide (BSTFA), containing 1% trimethylchlorosilane (TMC) was added to each vial and heated at 70˚C for 30 min. The samples were then evaporated to dryness under a gentle stream of nitrogen and re-dissolved with 50.0 μl of hexane and vortexed for 30 s to ensure a homogeneous solution.

To the wine-markers samples: 50.0 μl of N,O-bis(trimethyl)silyltrifluoroacetamide (BSTFA), containing 1% trimethylchlorosilane (TMC), 100.0 μl of DCM and 4.0 μl of pyridine were added to each vial and heated at 40˚C for 30 min. The samples were then evaporated to dryness under a gentle stream of nitrogen and re-dissolved with 50.0 μl of cyclohexane and vortexed for 30 s to ensure a homogeneous solution.

Eight μl of each sample were injected into the gas chromatograph (GC) coupled with a mass-selective detector (MSD). An analytical blank was also prepared by the same method with each batch of pottery samples.

**Gas chromatography/mass spectrometry (GC-MS).**    GC/MS measurements were carried out using a HP6890 GC equipped with a mass-selective detector (HP5973; electron multiplier potential 2 kV, filament current 0.35 mA, electron energy 70 eV, and the spectra were recorded every 1 s over the range m/z 50–800). Splitless injection was performed. 30 m, 0.32 mm ID 5% cross-linked phenylmethyl siloxane capillary column (HP-5) with a 0.25 mm film thickness was used for the separation. Helium was used as a carrier gas at a constant flow of 1.0 mL/s.

The injection temperature was 250˚C. The oven temperature was initially set at 70˚C (8 min. isothermic hold), ramped to 180˚C at 20˚C min−1, and then to 280˚C at 5˚C min−1 (8 min. isothermic hold). Peak assignments were based on comparisons with library spectra (NIST 17), spectra reported in the literature [24, 60, 61] and by comparison of retention times of reference standards.

**Ethics statement.** The excavations at Giv'ati Parking Lot at the City of David national park (Permit Nos. G-71/17, G-11/18 and G10/19), were conducted on behalf of the Israel Antiquities Authority and Tel-Aviv University. The pottery was scanned in 3-D and analyzed by the Computational Archaeology Laboratory at the Institute of Archaeology, Tel-Aviv University. The excavations in Area U within the area of the City of David National Park, were conducted by the Israel Antiquities Authority (IAA). The excavations were supported by the City of David Foundation. The excavation map (Fig 1) was prepared by the City of David expedition, headed by three of the authors, and the full copyright belongs to them. All necessary permits were obtained for the described study, which complied with all relevant regulations.

## Supporting information

**S1 Fig. A plate with the drawing of all storage jars found in Room C, Building 100, and included in this study (note that the Pithos mentioned in the text was not drawn).** This figure was produced by the Computational Archaeology Laboratory at the Institute of Archaeology, Tel-Aviv University.
(PDF)

**S2 Fig. A plate with the drawing of all storage jars found in Structure 17049 and included in this study.** The pottery was digitally scanned by Ortal Harush and Argita Gyerman-Levanon.
(PDF)

**S3 Fig. GC chromatograms of storage jars and pithoi found in Room C and included in this study.**
(PDF)

**S4 Fig. GC chromatograms of storage jars found in Building 17049 and included in this study.**
(PDF)

**S1 Table. Givati parking lot Rosette storage jars from Building 100.**
(PDF)

**S2 Table. RA results of all vessels included in this study.**
(PDF)

**S3 Table. Summary of RA results of all vessels included in this study.**
(XLSX)

## Acknowledgments

We would like to thank the restoration labs of the Israel Antiquities Authority (IAA) and Tel-Aviv University. We would like to thank The Mass Spectrometry Division of the Department of Chemistry in Bar Ilan University for helping with the analysis of the MS spectra.

## Author Contributions

**Conceptualization:** Yuval Gadot.

**Data curation:** Ayala Amir.

**Formal analysis:** Ayala Amir.

**Investigation:** Ayala Amir, Yiftah Shalev, Joe Uziel, Ortal Chalaf, Liora Freud.

**Methodology:** Ayala Amir.

**Project administration:** Ayala Amir, Joe Uziel, Yuval Gadot.

**Supervision:** Israel Finkelstein, Ronny Neumann, Yuval Gadot.

**Writing – original draft:** Ayala Amir, Joe Uziel, Yuval Gadot.

**Writing – review & editing:** Ayala Amir, Israel Finkelstein, Yiftah Shalev, Joe Uziel, Ronny Neumann, Yuval Gadot.

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
