## [Decision Letter · Decision Letter 0]

18 Jan 2022

PONE-D-21-38435Residue Analysis Evidence for Wine Enriched with Vanilla Consumed in Jerusalem on the Eve of the Babylonian Destruction in 586 BCEPLOS ONE

Dear Dr. Amir,

Thank you for submitting your manuscript to PLOS ONE. After careful consideration, we feel that it has merit but does not fully meet PLOS ONE’s publication criteria as it currently stands. Therefore, we invite you to submit a revised version of the manuscript that addresses the points raised during the review process.

We look forward to receiving your revised manuscript.

Kind regards,

Joseph Banoub, Ph,D., D. Sc.

Academic Editor

PLOS ONE

Journal Requirements:

3.  We note that Figure 1 in your submission contain map image which may be copyrighted. All PLOS content is published under the Creative Commons Attribution License (CC BY 4.0), which means that the manuscript, images, and Supporting Information files will be freely available online, and any third party is permitted to access, download, copy, distribute, and use these materials in any way, even commercially, with proper attribution. For these reasons, we cannot publish previously copyrighted maps or satellite images created using proprietary data, such as Google software (Google Maps, Street View, and Earth). For more information, see our copyright guidelines: http://journals.plos.org/plosone/s/licenses-and-copyright.

a) You may seek permission from the original copyright holder of Figure 1 to publish the content specifically under the CC BY 4.0 license.  

Reviewers' comments:

Reviewer's Responses to Questions

**Comments to the Author**

1. Is the manuscript technically sound, and do the data support the conclusions?

Reviewer #1: Yes

Reviewer #2: Yes

2. Has the statistical analysis been performed appropriately and rigorously? 

Reviewer #1: N/A

Reviewer #2: N/A

3. Have the authors made all data underlying the findings in their manuscript fully available?

Reviewer #1: Yes

Reviewer #2: Yes

4. Is the manuscript presented in an intelligible fashion and written in standard English?

Reviewer #1: Yes

Reviewer #2: Yes

5. Review Comments to the Author

Reviewer #1: The article submitted for expertise presents the results of residue analysis, based on the mass of the gas chromatograph spectrometer measurements (GC-MS), performed on ceramic storage jars

from the layer of Babylonian destruction in Jerusalem. 5 pots bear rosette stamp impressions indicating a link to the royal economy of the kingdom of Judah. Identification of the original

remains important for the understanding of many aspects including the nutrition, economics and international trade at this time. The study makes it possible to elucidate the contents of the jars, as well as the process of destruction of the buildings containing them. The study reveals that the jars were used alternately to store wine (flavored with vanilla) and olive oil. The results not only provide information on wine consumption habits at that time, but also reveal historical transregional South Arabian conflicts in relation to the trade in spices in particular.

The study presented here is very complete and very well presented. The results are well presented, commented on and thus well-supported conclusions are issued. The introduction as well as the conclusion would deserve to be improved a little in order to better position the study in relation to similar works and the conclusion must better highlight the nature and the prospects of the results obtained.

This article should be published after careful proofreading to correct minor errors.

Reviewer #2: This is an interesting scholarly article and ejoyable to read. It is well-written and I could only find very minor minor grammatical/terminological errors and these are added to illustrate:

1. In the "extraction and analysis procedures" there is not suffcient care given to PRECISION of the quantities employed. For examole: "5 ml of boron trifluoride," - has only a single significant figure: it could be 4 ml or 5 ml!! At least two if not three significant figures should be quoted. Also the correct SI

2.The correct usage should read: "an queous saturated solution..." NOT simply "a saturated solution of sodium

carbonate".

3. "Anhydrous" sodium sulfate was added to remove any residual water from the samples. The samples were filtered and were evaporated to dryness under a gentle stream of nitrogen and mild heating."

A personal comment: I found it facinating that the vanallin was so stable to oxidation over all of the years that the samples survived!

6. PLOS authors have the option to publish the peer review history of their article (what does this mean?). If published, this will include your full peer review and any attached files.

Reviewer #1: No

Reviewer #2: No

---

## [Author Response · Author response to Decision Letter 0]

26 Jan 2022

1. The manuscript was adapted to PLOS ONE's style requirements.

2. Our ethics statement was moved to the Methods section of the manuscript.

3. The excavation map (figure 1) was prepared by the City of David expedition, headed by three of the authors, and the full copyright belongs to them. This map hadn’t been published in any other journal. This information was included in the ethic statement section, and the appropriate credit was included in the figure legend as well.

4. The references were adapted to PLOS ONE's style requirements. Reference number 14 has been removed as it has not yet been published and there are other references in the article that address the relevant topic in this reference.

5. The introduction and the conclusions were edited in order to make sure they fit better the body of the article (results and discussion), following the comment made by the first reviewer. 

6. following the comments made by the second reviewer:

• Significant figures were added to the quantities mentioned in the methods section so two significant figures are quoted. (The quantities of reagents used in the extraction and silylation procedures are in large excess of the molecules with which they react in order to ensure complete extraction and derivatization. Therefore, there is no need for such accuracy).

• The words “an aqueous” were added to the relevant sentence.

• The word “Anhydrous” was added to the relevant sentence.

---

## [Editor Report · Decision Letter 1]

14 Mar 2022

Residue analysis evidence for wine enriched with vanilla consumed in Jerusalem on the eve of the Babylonian destruction in 586 BCE

PONE-D-21-38435R1

Dear Dr. Amir,

We’re pleased to inform you that your manuscript has been judged scientifically suitable for publication and will be formally accepted for publication once it meets all outstanding technical requirements.

Kind regards,

Joseph Banoub, Ph,D., D. Sc., FCIC, FRSC

Academic Editor

PLOS ONE

Additional Editor Comments (optional):

I am sorry for the delay in accepting this manuscript.

This was due to a technical error from PLOS.

All the best and take care
---

## [Editor Report · Acceptance letter]

21 Mar 2022

PONE-D-21-38435R1 

Residue analysis evidence for wine enriched with vanilla consumed in Jerusalem on the eve of the Babylonian destruction in 586 BCE 

Dear Dr. Amir:

I'm pleased to inform you that your manuscript has been deemed suitable for publication in PLOS ONE. Congratulations! Your manuscript is now with our production department. 

Kind regards, 

on behalf of

Dr. Joseph Banoub 

Academic Editor

PLOS ONE